# Short-Term Mediterranean Dietary Intervention Reduces Plasma Trimethylamine-N-Oxide Levels in Healthy Individuals

**DOI:** 10.3390/nu17193135

**Published:** 2025-09-30

**Authors:** Melike Şeyma Deniz, Murat Baş

**Affiliations:** 1Department of Nutrition and Dietetics, Institute of Health Sciences, Acibadem Mehmet Ali Aydınlar University, 34752 Istanbul, Turkey; melike.deniz@istinye.edu.tr; 2Department of Nutrition and Dietetics, Faculty of Health Sciences, İstinye University, 34010 Istanbul, Turkey; 3Department of Nutrition and Dietetics, Faculty of Health Sciences, Acıbadem Mehmet Ali Aydınlar University, 34752 Istanbul, Turkey

**Keywords:** mediterranean diet, TMAO, cardiovascular diseases, MEDAS, microbiota

## Abstract

**Objectives**: This study aimed to evaluate the association between adherence to the Mediterranean diet and blood trimethylamine-N-oxide (TMAO) levels. **Methods**: This randomized clinical trial enrolled 53 healthy adults with normal or overweight body mass index (BMI) who were recruited from a cardiology outpatient clinic in Istanbul, Turkey. Dietary patterns and Mediterranean diet adherence (assessed using the MEDAS) were evaluated alongside anthropometric and biochemical parameters, including fasting glucose, total cholesterol, high-density lipoprotein (HDL), low-density lipoprotein (LDL), triglycerides, alanine aminotransferase (ALT), and aspartate aminotransferase (AST). Twenty individuals with low adherence underwent a 4-week Mediterranean dietary intervention with daily dietary monitoring. To assess changes, pre- and postintervention data were compared. **Results**: The results revealed that individuals adhering to the Mediterranean diet exhibited significantly lower blood TMAO levels (*p* < 0.001). In males, total cholesterol, LDL, triglyceride, and ALT levels significantly decreased compared with those at baseline (*p* < 0.05), whereas fasting blood glucose, HDL, and AST levels showed no significant changes (*p* > 0.05). In females, only blood TMAO levels exhibited significant reduction, with no other biochemical parameters indicating significant differences (*p* > 0.05). Additionally, males demonstrated significant improvements in anthropometric measures, including weight, BMI, fat mass, muscle mass, waist, and hip circumference, compared with pre-intervention values (*p* < 0.05), whereas females exhibited no significant changes in these measures (*p* > 0.05). **Conclusions**: Our findings demonstrate that implementing the Mediterranean diet in individuals with initially low adherence causes significantly reduced blood TMAO levels even within a short intervention period of 4 weeks.

## 1. Introduction

Characterized by its diversity, the Mediterranean diet reflects the common culinary traditions of Mediterranean countries, including high contents of olive oil and olives, vegetables, fruits, unrefined grains, legumes, and nuts; moderate intake of fish and dairy products; and low meat and meat product consumption, and it is a term expressed not only as a diet but as a way of life [1]. Several studies have suggested that the Mediterranean diet can protect against cardiovascular diseases, stroke, obesity, diabetes, hypertension, various types of cancer, allergies, and neurodegenerative diseases (Alzheimer’s and Parkinson’s diseases), and the Mediterranean diet is associated with longevity [2,3,4,5]. Conversely, there is growing evidence that the gut microbiota plays a crucial role in health and disease development. Recently, studies have aimed at elucidating the association between the gut microbiota and cardiovascular risk. These studies have suggested that some bile acids, short-chain fatty acids, and intestinal metabolites (trimethylamine-N-oxide [TMAO]) are involved in the development and progression of cardiovascular risk [6].

Wang et al. [7] reported that a metabolic pathway exists wherein phosphatidylcholine, choline, betaine, and L-carnitine are converted to trimethylamine (TMA) in the intestine, with majority of the TMA ingested or produced in the intestine being rapidly absorbed into the portal circulation by passive diffusion and subsequently oxidized to TMAO by the action of the flavin-containing monooxygenases FMO3 and FMO1. Diets are significantly involved in TMAO production as they provide the nutritional precursors for generating TMA and TMAO [6]. TMAO has been implicated in cardiovascular and metabolic diseases, in part through its promotion of oxidative stress and chronic inflammation. Mechanistically, TMAO may increase reactive oxygen species, suppress antioxidant defenses, and activate inflammatory pathways, including NF-κB signaling and cytokine production. Elevated TMAO levels have been associated with higher circulating C-reactive protein (CRP) in humans, supporting its role in systemic inflammation [8,9]. Dietary interventions, such as the Mediterranean diet, reduce circulating TMAO levels, suggesting a potential pathway by which diet may mitigate oxidative stress and inflammation and contribute to cardiovascular protection [10].

The Mediterranean diet is associated with a reduced risk of chronic diseases and may counteract the pro-inflammatory effects of elevated TMAO production. Although the impact of the Mediterranean diet on circulating TMAO levels remains unclear, some studies have suggested that it can lower TMAO concentrations. For example, De Filippis et al. [11] demonstrated that adherence to the Mediterranean diet was significantly inversely correlated with urine TMAO levels in a cohort of vegans, vegetarians, and omnivores in Italy. Similarly, Kühn et al. [12] reported that high consumption of Mediterranean diet-based foods, including vegetables, fruits, and nuts, was associated with the lowest circulating TMAO levels. Moreover, a randomized controlled trial has revealed that adopting a Mediterranean-style eating pattern with low (but not moderate) intake of unprocessed lean red meat significantly reduced fasting serum TMAO levels in adults with overweight or obesity. This finding supports the idea that lower red meat intake within a Mediterranean diet may effectively reduce TMAO levels [12]. Studies on the association between urine and/or plasma TMAO levels and the consumption of TMA-rich foods of animal origin have indicated that TMAO production widely varies among individuals [13]. Several factors, including diet type, liver FMO activity, and kidney function, determine TMAO levels. Genetic factors are believed to have a minor impact in healthy individuals, and the interrelationships between TMA and/or TMAO precursor foods and gut microbiota are more effective on TMAO variation [14]. The idea that high blood TMAO levels can be used as a marker for several diseases, particularly cardiovascular diseases, necessitated determining potential approaches to lower TMAO levels. One of these approaches is reducing the intake of TMAO precursors, including phosphatidylcholine, choline, betaine, and L-carnitine, in nutritional programs [14]. Several studies have reported the benefits of Mediterranean diet in cardiovascular disease prevention and management [2,15]. In this case, individuals who follow a Mediterranean diet are anticipated to have lower blood TMAO levels and a reduced risk of cardiovascular disease. This study aimed to evaluate the association between Mediterranean diet adherence and blood TMAO levels in adults. It was hypothesized that circulating TMAO levels would vary according to dietary habits and that individuals with higher adherence to the Mediterranean diet, characterized by lower intake of red meat and full-fat dairy products, would exhibit lower TMAO levels owing to the reduced concentrations of TMAO precursors, including L-carnitine and phosphatidylcholine.

## 2. Materials and Methods

### 2.1. Participants and Study Design

This randomized clinical trial was conducted with healthy volunteers aged 19–65 years who were recruited from a hospital cardiology outpatient clinic in Istanbul, Turkey from June to August 2021. This study was conducted in accordance with the Declaration of Helsinki and approved by the Medical Research Ethics Committee of Acıbadem Mehmet Ali Aydınlar University (decision number, 2019-5/13; date, 3 July 2019 meeting number, 2019/5). Written informed consent was obtained from all participants who voluntarily agreed to participate in this study. The following were the exclusion criteria: females who were postmenopausal or had experienced amenorrhea for ≥3 months; individuals of both sexes using hormonal contraceptives; those who had taken probiotics or antibiotics within the past 2 months; those adhering to specific diets, including vegan or vegetarian; those receiving vitamin, mineral, or antioxidant supplementation; and those using nonsteroidal medications, diuretics, or laxatives.

Based on an estimated outpatient flow of 400 individuals over the 3-month recruitment period, a power analysis was conducted using G*Power (version 3.1.9.6; Heinrich-Heine-Universität Düsseldorf, Düsseldorf, Germany). Assuming a medium-to-large effect size (Cohen’s d = 0.7), a significance level of 0.05, and a desired power of 80%, the minimum required sample size for a paired-samples *t*-test was calculated as 19 participants. Accordingly, to meet and slightly exceed this requirement, 20 individuals with low Mediterranean diet adherence were included in the intervention group.

Among the patients who applied to the cardiology outpatient clinic within 3 months, 53 volunteered to participate in this study. A demographic characteristics questionnaire, food frequency questionnaire, and the Mediterranean Diet Adherence Screener (MEDAS) scale were administered to the participants, and a 7-day food record was collected from them. The researcher collected anthropometric data from participants during their hospital visit for extracting blood samples, including fasting blood glucose, total cholesterol, low-density lipoprotein (LDL) cholesterol, high-density lipoprotein (HDL) cholesterol, triglyceride, alanine aminotransferase (ALT), and aspartate aminotransferase (AST), and TMAO levels. Using the MEDAS scale, 37 individuals with low Mediterranean diet adherence were identified. These individuals were informed about the diet phase of the study, and 21 of them agreed to participate in the Mediterranean diet program. At this stage, one individual did not continue the program, and the Mediterranean diet program was planned with 20 participants. The remaining participants who did not take part in the dietary intervention were excluded from the follow-up and did not serve as a control group. The group who started the diet was provided face-to-face training regarding the Mediterranean diet, a 1-week sample Mediterranean diet planned according to the daily energy requirements of the individuals, and an information brochure about the Mediterranean diet. The dietary intervention was designed to be isocaloric, ensuring that each participant’s energy intake matched their estimated requirements to maintain weight.

For the scheduled weekly Mediterranean diet program, the Harris–Benedict equation was used for calculating the energy requirements of the participants, and the weekly program was prepared on the basis of the Mediterranean pyramid [16]. One-month food records were collected from the participants. After 1 day, food consumption was monitored via the online communication application. Weekly evaluations were conducted, and changes and guidance were made when necessary. After 1 month, the participants returned to the hospital, trained nurses extracted blood samples, and the researcher took their anthropometric measurements.

### 2.2. Anthropometric Measurements

After measuring the body weight and height of the participants in light clothing and on an empty stomach following standard protocol, body composition analysis was performed using the bioelectrical impedance analysis (BIA) method; subsequently, body weight (kg), lean body mass (kg) and percentage, body water amount (lt) and percentage, basal metabolic rate (kcal), and body mass index (BMI) (kg/m^2^) were recorded. The device (Tanita Corporation, Tokyo, Japan) was used for BIA measurement. The height of the participants was measured using a Seca brand height meter, with the feet side by side and the head in the Frankfurt plane. Waist circumference was measured using a tape measure passing through the midpoint of the lateral iliac prominences and the lowest rib while the individual was standing. Hip circumference was measured from the highest point, standing on the side of the individual, and the waist/hip ratio was calculated and recorded [17].

### 2.3. Food Intake

The basic eating habits questions in the demographic questionnaire, the food frequency questionnaire, and the 7-day food record were administered to determine the nutritional habits of the participants. Daily consumptions were investigated using the Computer-Aided Nutrition Program, Nutrition Information Systems Package Program (BEBIS) [18].

### 2.4. Blood Parameters

The physician evaluated all participants. Trained nurses performed blood sample extraction for fasting blood glucose, total cholesterol, HDL cholesterol, LDL cholesterol, triglyceride, ALT, AST, and blood TMAO levels. Blood samples were collected in three tubes, including one tube for fasting blood glucose evaluation; one tube for total cholesterol, HDL cholesterol, LDL cholesterol, triglyceride, ALT, and AST evaluations; and one for TMAO level evaluation.

Blood TMAO was analyzed using a BT LAB-Human Trimethylamine-N-Oxide ELISA kit. Biochemical parameters, other than TMAO, were evaluated in Labmed, the laboratory of the Acıbadem Healthcare Group, and the TMAO level was maintained at the Opakgen Laboratory under appropriate conditions (−20 °C) for evaluation.

### 2.5. Statistical Analysis

Daily energy and nutrient intakes were analyzed using the Nutrition Information System (BEBIS) 7.2 program. Statistical analysis was performed using the IBM SPSS 20.0 (SPSS Inc., Chicago, IL, USA). The normal distribution test was evaluated using the Kolmogorov–Smirnov test. Normally distributed numerical variables were presented as means ± standard deviations, nonnormally distributed numerical variables were expressed as medians (25th–75th quartiles), and categorical variables were expressed as frequencies (percentiles). In the case of two groups, Student’s *t*-test was used for numerical variables with normal distribution and the Mann–Whitney U test for nonnormally distributed numerical variables. Comparisons between the two dependent groups were conducted using the paired-samples *t*-test for normally distributed numerical variables, while the Wilcoxon signed-rank test was applied for variables that deviated from normality. Moreover, Pearson’s chi-squared and Fisher’s exact tests were performed to determine associations between categorical variables. To investigate the relationship between two numerical variables, correlation analysis was employed. *p* < 0.05 was considered statistically significant.

## 3. Results

This study encompassed 53 participants, including 30 females and 23 males. The mean age of the entire cohort was 38.89 ± 8.86 years. Specifically, the mean age of the female and male participants was 41.00 ± 8.40 and 36.13 ± 8.86 years, respectively. Regarding marital status, 56.6% of the participants were married, whereas 43.4% were single. Regarding education status, 79.2%, 17%, and 3.8% of the participants were university, high school, and primary school graduates, respectively. Although 69.8% of the participants reported having no diagnosed illness, 30.2% reported having a diagnosed medical condition. Moreover, 92.5% of the participants stated that they were not taking any medications. Regarding alcohol consumption, 50.9% of the participants stated consuming alcohol, whereas 49.1% did not. Regarding smoking status, 21.2% of the participants reported being smokers, whereas 78.8% stated that they did not smoke. The general characteristics of the participants are presented in Table 1.

The mean body weight of the female participants was 64.39 ± 9.83 kg, whereas that of the male participants was 77.24 ± 7.24 kg. The BMI values were 24.73 ± 3.51 and 24.83 ± 2.75 kg/m^2^ for females and males, respectively. In females, the average body fat, muscle, and water percentages were 26.52% ± 8.37%, 70.52% ± 6.66%, and 49.83% ± 4.16%, respectively. In males, the average body fat, muscle, and water percentages were 20.88% ± 9.48%, 77.16% ± 4.36%, and 56.88% ± 3.20%, respectively. The mean waist circumference, hip circumference, and waist-to-hip ratio in females were 77.06 ± 10.12 cm, 97.06 ± 6.62 cm, and 0.78 ± 0.05, respectively, whereas those in males were 87.91 ± 6.59 cm, 98.56 ± 4.05 cm, and 0.88 ± 0.03, respectively. The participants’ anthropometric measurements are shown in Table 2.

The mean daily energy intake for females and males was 1240.75 ± 341.82 and 1697.12 ± 402.62 kcal, respectively. In females, carbohydrate, protein, and fat intakes were 107.09 ± 47.27 g, 51.96 (46.85–57.72) g, and 65.68 ± 17.79 g, respectively, whereas those in males were 156.81 ± 51.61 g, 80.98 (62.13–94.93) g, and 79.94 ± 20.56 g, respectively. The average intakes of polyunsaturated, saturated, and monounsaturated fatty acids among all participants were 16.70 ± 5.45 g, 25.66 ± 7.92 g, and 24.29 ± 7.57 g, respectively. The mean daily fiber intake was 14.53 ± 4.70 g in females, 17.16 ± 6.78 g in males, and 15.67 ± 5.79 g overall. The mean daily cholesterol intake in females was 289.42 ± 132.17 mg, whereas that in males was 350.11 ± 125.30 mg. These values were calculated on the basis of the 7-day dietary records collected from the participants before the intervention. The energy, protein, carbohydrate, fat, vitamin, mineral, fiber, and cholesterol intakes of the participants are presented in Table 3.

The MEDAS scale was administered to the participants on their first visit, and their Mediterranean diet adherence was evaluated. The grouping of the participants according to the MEDAS scale is presented in Table 4. Among the participants, 69.8%, 11.3%, and 18.9% had scores of <7, 7–8, and ≥9, respectively. It was observed that 70% of the female participants and 69.6% of the male participants exhibited scores of <7.

The anthropometric measurements of the intervention group of 20 participants who were subjected to the Mediterranean diet at the beginning and at the end of the 4-week dietary intervention are provided in Table 5. Males demonstrated statistically significant difference in measurements, including weight, BMI, fat mass, waist circumference, and hip circumference (*p* < 0.05). In females, no significant difference was noted in anthropometric measurements at the end of the dietary intervention (*p* > 0.05).

The biochemical parameters of the intervention group of 20 participants included in the Mediterranean diet at the beginning and at the end of the 4-week dietary intervention are shown in Table 6. The blood TMAO level significantly decreased following the dietary intervention in both female and male participants (*p* < 0.05). Although the total cholesterol, LDL cholesterol, triglyceride, and ALT levels significantly differed in males (*p* < 0.05), no significant difference was observed in females (*p* > 0.05).

## 4. Discussion

Recent studies have suggested that dietary food intake and subsequent metabolism of nutrients by the intestinal microbiota are involved in the pathogenesis of cardiometabolic diseases [19]. Red and processed meats, egg yolks, and other animal-based foods are rich in choline, phosphatidylcholine, carnitine, and trimethylamine, and when metabolized by the gut microbiota, TMAO is formed, which has been demonstrated to correlate with cardiovascular disease [20]. The physiological and pathophysiological effects of TMAO are much debated [19]. While TMAO is implicated in cardiovascular and renal pathology, it also fulfills important physiological roles, including acting as an osmolyte and modulating immune responses [21]. Some studies suggest that TMAO may exert protective effects under certain conditions, such as reducing diastolic dysfunction in hypertensive rats [21] and enhancing antitumor immunity in cancer models [22]. Chronic suppression could theoretically impair these beneficial functions; however, direct evidence of harm from long-term TMAO reduction is currently lacking in animal studies [23,24]. Human data on adverse effects from sustained low TMAO levels remain limited, and inter-individual differences in TMAO metabolism may influence outcomes [25]. Overall, current research supports the durability and therapeutic benefit of long-term TMAO suppression for cardiovascular and renal disease, with no clear evidence of major adverse effects. Nevertheless, the full spectrum of TMAO’s physiological functions—particularly in humans—remains incompletely understood, and further long-term studies are warranted to ensure that chronic TMAO reduction does not inadvertently compromise its beneficial roles [24].

Although different variations are considered, including gut microbiota composition, liver enzyme activity, and methylamine excretion rate, it is widely accepted that circulating TMAO levels primarily depend on dietary pattern [25]. The effects of various dietary interventions on TMAO levels has been investigated less despite the increasing number of studies on the negative effects of high TMAO levels on health [26,27].

The intake of specific TMAO precursor-rich food, including eggs, beef, and fish, is widely known to increase blood and urine levels of TMAO [14]. For example, Miller et al. reported a dose–response relationship between the consumption of eggs, a source of phosphatidylcholine, and both plasma and urine TMAO levels in six healthy volunteers [28]. Fish and seafood consumption has been shown to be associated with high TMAO levels [29]. In the cross-sectional KarMeN study [30], fish consumption in an adult population in Germany was associated with increased TMAO levels in both plasma and urine samples. This finding is noteworthy, as it is well known that TMAO-rich fish and seafood have plasma lipid-lowering and anti-inflammatory effects and protective effects from cardiovascular diseases, mainly due to the presence of polyunsaturated fatty acids. In humans, plasma TMAO levels are approximately 50-fold higher following fish consumption than those following egg and beef intake [31]. It seems paradoxical that fish and seafood contain high levels of both polyunsaturated fatty acids and TMAO, which is associated with major cardiovascular events and mortality; however, TMAO in fish likely comes from natural food sources and tissue accumulation rather than a metabolic product [29]. A study evaluating the effects of an 8-week vegan dietary intervention on TMAO levels in adults with obesity or dysglycemia revealed that the plasma TMAO level decreased in the first week of the vegan diet, and this effect persisted in the 8th week [32]. Studies investigating the effects of omnivorous and vegetarian/vegan diets on TMAO levels concluded that those following a vegetarian/vegan diet had lower TMAO levels [33]. Considering the effects of TMA or TMAO precursor metabolites on blood or urine TMAO levels, it is anticipated that those who follow a diet rich in animal-derived foods will have higher TMAO levels. The fact that vegetarian/vegan dietary interventions decrease TMAO levels confirms this expectation. Plant-based diets have been demonstrated to provide nutritional advantages over omnivorous diets, with benefits for cardiovascular health, blood pressure, and plasma lipids. A fiber-rich Mediterranean diet is associated with a reduced risk of chronic disease, and increased TMAO formation may counteract the pro-inflammatory effects. Moreover, high fruit, nut, and vegetable consumption in the Mediterranean diet is associated with the lowest plasma TMAO levels, whereas high consumption of animal-based foods, including red meat or fish, is associated with the highest plasma TMAO levels [34]. Analyzing the studies investigating the effects of the Mediterranean diet on TMAO levels yielded varying results. Griffin et al. investigated 115 participants following a Mediterranean diet for 6 months and observed no significant changes in TMA or TMAO precursors, suggesting that the 6-month extensive dietary intervention is insufficient for reducing TMAO levels [35]. Conversely, Barrea et al., in their study investigating the association between TMAO and the Mediterranean diet in healthy and normal-weight individuals, reported that males had lower adherence to the Mediterranean diet, subsequently increasing their TMAO levels [34]. Although 4 weeks may appear brief in our study, the reduction in blood TMAO levels in both male and female participants aligns with current evidence confirming that the gut microbiome can rapidly respond to an altered diet [36]. Our findings suggest that the reduction in plasma TMAO levels observed in our study is primarily attributable to dietary composition rather than weight loss per se. This interpretation is consistent with evidence from a longitudinal cohort of US men, where detailed dietary assessments combined with gut microbiome profiling and metabolomic analyses demonstrated that specific dietary patterns—particularly higher intake of red meat and choline—significantly influenced circulating TMAO concentrations through modulation of gut microbial composition and function [37]. These results underscore the importance of integrating microbiota profiling and metabolomics in future research to further elucidate the mechanisms by which diet regulates TMAO metabolism. In our study, although males demonstrated significant positive changes in anthropometric measurements, no significant difference in anthropometric measurements were noted in females; however, blood TMAO levels significantly decreased in both groups. Several factors may account for these disparities. Physiologically, variations in sex hormones, body fat distribution, and hepatic enzyme activity can influence both metabolic regulation and gut microbiota composition, thereby altering dietary responses [38]. In addition, behavioral aspects such as dietary preferences, adherence to the intervention, and motivation may contribute. For example, women often report higher carbohydrate intake, frequently favoring low-fiber sources such as rice and potatoes, which may attenuate the lipid-lowering effects of the Mediterranean diet [39]. In contrast, men in our study reported higher baseline consumption of animal-derived foods; thus, the transition to a diet richer in plant-based proteins, legumes, and nuts may have resulted in more substantial metabolic and anthropometric improvements. The absence of a structured physical activity program may also explain why HDL concentrations did not improve in men despite reductions in total cholesterol, LDL, and triglycerides. Beyond biological factors, gender-related roles and social determinants may shape food choices, cooking practices, and dietary adherence, influencing the effectiveness of nutritional interventions [40]. These findings suggest that sex is an important modifier of dietary response and highlight the need for future research to consider sex- and gender-specific strategies when designing dietary interventions.

Mediterranean diet adherence has also been associated with improvements in other metabolic health parameters. It has been linked to reduced risk of chronic conditions, including obesity, type 2 diabetes, and cardiovascular diseases, primarily through improvements in lipid profile, glucose metabolism, and blood pressure [41]. Although oxidative stress and inflammatory markers were not measured in the present study, TMAO is known to influence these pathways through multiple mechanisms, including increased reactive oxygen species production, suppression of antioxidant defenses, and activation of inflammatory signaling. In parallel, a 2025 meta-analysis of 65 studies reported that adherence to the Mediterranean diet was weakly associated with reductions in oxidative stress markers (e.g., MDA, 8OHdG) and inflammation markers (e.g., CRP, IL-6) in healthy individuals. While these reductions were generally not statistically significant, most estimates indicated a trend toward improvement, suggesting that dietary interventions may help lower early-stage oxidative stress and inflammation [10]. Taken together, these findings indicate that future studies incorporating direct assessment of oxidative stress and inflammatory markers would be valuable to elucidate the combined effects of TMAO reduction and Mediterranean diet adherence on cardiometabolic risk.

Tzima et al. [42] reported that individuals with high adherence exhibited better insulin sensitivity and lower blood pressure. Similarly, Sangouni et al. [43] observed that Mediterranean diet adherence improved AST and gamma-glutamyl transferase levels; however, no significant effect was observed on ALT. In our study, we observed sex-specific differences in the response to the intervention. Males experienced significantly reduced total cholesterol, LDL, triglyceride, and ALT levels, whereas females exhibited no such changes. Furthermore, anthropometric parameters, including weight, BMI, and fat mass, significantly improved in males but remained unchanged in females. These results suggest that the Mediterranean diet does not negatively affect the metabolic profile and can provide more rapid benefits in males [44]. Previous studies have suggested that longer intervention durations and complementary strategies, such as energy restriction or physical activity, are necessary to observe more pronounced effects, particularly in females [45]. Esposito et al. [46], in a systematic review of 16 randomized controlled trials, reported greater weight loss with the Mediterranean diet, but only when accompanied by caloric restriction or physical activity and when the duration exceeded 6 months. Our findings support the growing body of evidence indicating that adherence to a Mediterranean dietary pattern positively affects cardiometabolic health beyond TMAO reduction.

Sex-specific differences in response to dietary intervention highlight the significance of tailored nutrition strategies. Future studies should focus on the long-term impact of Mediterranean diet adherence on TMAO metabolism, liver function, and overall metabolic health in diverse populations.

### Strengths and Limitations

One of the main strengths of this study is its prospective design and the use of a structured Mediterranean dietary intervention tailored to individual energy needs. Moreover, the daily follow-up of participants and dietary compliance monitoring through an online communication platform enhanced the accuracy of dietary adherence assessment. The use of validated tools, including the MEDAS scale, and standardized biochemical analyses further strengthens the internal validity of this study. Despite its strengths, this study had several limitations. A limitation of the current study is the absence of a control group. The within-subject design was chosen for practical reasons, but future research incorporating a matched non-intervention group or a crossover design would strengthen internal validity and allow for more definitive conclusions regarding the effects of the Mediterranean diet. Second, the relatively small sample size, especially within the intervention group, limited the generalizability of the findings. Moreover, participants reported all meals and snacks daily via a digital messaging platform. I monitored these reports and provided individualized feedback to support adherence and guide adjustments. Meal-by-meal progress was tracked throughout the intervention. As reporting relied on participant self-report, unrecorded intake could not be verified, representing a limitation of the monitoring approach. Lastly, the lack of gut microbiota profiling prevented the exploration of potential mechanisms behind individual variations in TMAO response. Future studies incorporating microbiome analyses could provide deeper insights into these mechanisms and the role of gut microbial composition in mediating dietary effects.

## 5. Conclusions

This study demonstrates that a short-term (4-week) Mediterranean diet intervention tailored to individual energy needs can significantly decrease plasma TMAO levels, a marker associated with cardiovascular risk, in both males and females. The observed decrease occurred independent of weight loss, suggesting that the diet’s composition, rather than weight change, plays a major role in modulating TMAO levels. These findings support the practical implementation of the Mediterranean diet as a nonpharmacological strategy for reducing cardiometabolic risk by modulating gut-derived metabolites. The results may guide clinicians and dietitians in dietary counseling, particularly for individuals with low adherence to healthy eating patterns. Future research should focus on the long-term impact of dietary patterns on TMAO metabolism and related health outcomes.

## Figures and Tables

**Table 1 nutrients-17-03135-t001:** General characteristics of the participants.

	Females (n = 30)		Males (n = 23)		Total (n = 53)	
Age (X ± SD) (years)	41.00 ± 8.40		36.13 ± 8.86		38.89 ± 8.86	
	**N**	**%**	**N**	**%**	**N**	**%**
Marital status						
Single	15	50	8	34.8	23	43.4
Married	15	50	15	65.2	30	56.6
Education level						
Low	2	6.7	-	-	2	3.8
Intermediate	4	13.3	5	21.7	9	17.0
High	24	80.0	18	78.3	42	79.2
Any chronic disease						
Yes	11	36.7	5	21.7	16	30.2
No	19	63.3	18	78.3	37	69.8
Using any medication						
Yes	3	10.0	1	4.3	4	7.5
No	27	90.0	22	95.7	49	92.5

X ± SD, mean ± standard deviation.

**Table 2 nutrients-17-03135-t002:** Anthropometric measurements of the participants.

	Females (n = 30)	Males (n = 23)	Total (n = 53)
**Values**			
	**X ± SD**	**X ± SD**	**X ± SD**
Weight (kg)	64.39 ± 9.83	77.24 ± 7.24	69.96 ± 10.84
BMI (kg/m^2^)	24.73 ± 3.51	24.83 ± 2.75	24.77 ± 3.17
Fat mass (kg)	17.04 ± 6.75	14.66 ± 4.40	16 ± 5.92
Fat (%)	26.52 ± 8.37	20.88 ± 9.48	24.07 ± 9.22
Muscle mass (kg)	53.75 ± 8.98	59.44 ± 4.82	50.56 ± 10.79
Muscle (%)	70.52 ± 6.66	77.16 ± 4.36	73.40 ± 6.62
Water (kg)	31.74 ± 2.88	43.77 ± 2.74	36.96 ± 6.63
Water (%)	49.83 ± 4.16	56.88 ± 3.20	52.89 ± 5.14
Waist (cm)	77.06 ± 10.12	87.91 ± 6,59	81.77 ± 10.24
Hip (cm)	97.06 ± 6.62	98.56 ± 4.05	97.71 ± 5.65
Waist/Hip	0.78 ± 0.05	0.88 ± 0.03	0.83 ± 0.06

X ± SD, mean ± standard deviation.

**Table 3 nutrients-17-03135-t003:** Energy, macronutrient, and micronutrient intakes.

	Females	Males	Total	*p*
Values				
	(n = 30)	(n = 23)	(n = 53)	
Energy (kcal)	1240.75 ± 341.82	1697.12 ± 402.62	1438 ± 431.15	<0.001 ^a^
CHO (g)	107.09 ± 47.27	156.81 ± 51.61	128.67 ± 54.70	0.001 ^a^
Protein (g)	51.96 (46.85–57.72)	80.98 (62.13–94.93)	56.35 (50.86–79.86)	<0.001 ^b^
Fat (g)	65.68 ± 17.79	79.94 ± 20.56	71.87 ± 20.16	0.009 ^a^
PUFA (g)	15.37 ± 4.38	18.43 ± 6.27	16.70 ± 5.45	0.041 ^a^
SFA (g)	23.60 ± 8.09	28.35 ± 6.98	25.66 ± 7.92	0.029 ^a^
MUFA (g)	22.10 ± 6.16	27.15 ± 8.38	24.29 ± 7.57	0.015 ^a^
Omega 3 (g)	1.76 ± 0.54	2.44 ± 1.09	2.05 ± 0.89	0.011 ^a^
Omega 6 (g)	13.51 ± 4.29	15.67 ± 5.25	14.45 ± 4.80	0.105 ^a^
Thiamine (mg)	0.56 ± 0.12	0.77 ± 0.25	0.65 ± 0.22	0.001 ^a^
Niacin (mg)	9.18 ± 2.30	14.18 ± 4.55	11.35 ± 4.24	<0.001 ^a^
Vit B12 (mg)	3.87 ± 1.52	5.78 ± 2.10	4.70 ± 2.02	<0.001 ^a^
Vit E (mg)	13.99 ± 3.96	15.56 ± 5.06	14.67 ± 4.49	0.209 ^a^
Fiber (g)	14.53 ± 4.70	17.16 ± 6.78	15.67 ± 5.79	0.102 ^a^
Cholesterol (mg)	289.42 ± 132.17	350.11 ± 125.30	315.76 ± 131.55	0.096 ^a^
Vit A (RE)	908.68 (638.35–1191.50)	705.07 (614.02–976.10)	801.27 (635.27–1100.70)	0.127 ^b^
Vit B6 (mg)	0.93 ± 0.19	1.25 ± 0.33	1.07 ± 0.30	<0.001 ^a^
Riboflavin (mg)	0.96 ± 0.20	1.26 ± 0.35	1.09 ± 0.31	<0.001 ^a^
Vit C (mcg)	68.95 (47.56–84.89)	55.69 (43.48–83.26)	63.89 (45.06–83.13)	0.419 ^b^

a, Independent groups *t*-test (mean ± standard deviation); b, Mann–Whitney U test (median [Q1–Q3]). Abbreviations: CHO, carbohydrate; PUFA, polyunsaturated fatty acid; SFA, saturated fatty acid; MUFA, monounsaturated fatty acid.

**Table 4 nutrients-17-03135-t004:** MEDAS scores.

Groups (Points)	Females		Males		Total		*p*
	(n = 30)		(n = 23)		(n = 53)		
	S	%	S	%	S	%	
<7	21	70.0	16	69.6	37	69.8	
7–8	4	13.3	2	8.7	6	11.3	0.755 ^c^
>9	5	16.7	5	21.7	10	18.9	

%, column percentage; c, Chi-square test.

**Table 5 nutrients-17-03135-t005:** Anthropometric measurements.

		Males (n = 10)			Females (n = 10)	
	**First**	**Second**	** *p* **	**First**	**Second**	** *p* **
Weight (kg)	79.20 ± 8.41	76.57 ± 8.75	0.001 *	69.65 ± 8.57	68.43 ± 9.40	0.095
BMI (kg/m^2^)	25.12 ± 2.17	24.28 ± 2.34	0.001 *	26.92 ± 2.37	26.43 ± 2.58	0.071
Fat mass (kg)	15.79 ± 3.95	14.70 ± 4.60	0.021 *	21.91 ± 5.59	21.60 ± 5.58	0.425
Fat (%)	19.70 ± 3.17	18.85 ± 4.06	0.085	33.50 ± 7.47	31.19 ± 4.30	0.405
Muscle mass (kg)	60.24 ± 4.91	58.79 ± 4.69	0.001 *	45.34 ± 4.18	44.44 ± 4.51	0.190
Muscle (%)	76.28 ± 3.02	77.10 ± 3.86	0.082	65.42 ± 4.45	65.29 ± 4.07	0.809
Water mass (kg)	44.93 ± 2.62	43.96 ± 2.53	<0.001 *	33.16 ± 3.20	32.31 ± 1.86	0.287
Water (%)	57.00 ± 3.11	57.77 ± 3.66	0.040 *	47.89 ± 3.93	47.67 ± 3.79	0.816
Waist (cm)	90.2 ± 6.19	88.4 ± 7.33	0.014 *	84.20 ± 8.43	83.60 ± 8.50	0.217
Hip (cm)	100.00 ± 3.33	99.10 ± 4.22	0.068	101.90 ± 5.38	101.60 ± 5.66	0.434
Waist/Hip	0.89 ± 0.03	0.88 ± 0.03	0.019 *	0.82 ± 0.03	0.81 ± 0.04	0.443

* *t*-test in dependent groups. Abbreviation: BMI, body mass index.

**Table 6 nutrients-17-03135-t006:** Biochemical parameters.

		Males (n = 10)			Females (n = 10)	
	**First**	**Second**	** *p* **	**First**	**Second**	** *p* **
TMAO ^b^	14.79 (1.04–29.91)	8.12 (5.92–25.79)	0.005 *	15.99 (12.56–19.86)	6.50 (5.84–6.85)	0.005 *
FBS ^a^	94.60 ± 5.14	94.50 ± 4.94	0.963	95.00 ± 9.75	94.50 ± 4.94	0.714
Total cholesterol ^b^	221.5 (203.25–230.5)	186.5 (165.25–208)	0.009 *	211.5 (189–235)	214 (190–231)	0.541
HDL cholesterol ^a^	48.20 ± 13.23	46.20 ± 6.77	0.549	61.60 ± 10.91	62.10 ± 12.01	0.749
LDL cholesterol ^a^	151.40 ± 23.04	132.10 ± 27.35	0.035 *	129.50 ± 45.62	133.70 ± 44.98	0.474
Triglyceride ^b^	115.5 (76.5–203)	82 (56–169.75)	0.013 *	101.5 (62–130.5)	85 (61–110)	0.414
ALT ^b^	41.5 (29–55)	31.5 (28–33.75)	0.022 *	29.5 (24.25–35.75)	25 (21–29.5)	0.139
AST ^a^	20.60 ± 5.44	18.80 ± 4.70	0.457	18.70 ± 4.00	17.90 ± 4.60	0.621

a, paired-samples *t*-test (mean ± standard deviation); b, Wilcoxon signed-rank test (median [Q1–Q3]) * *p* < 0.05. Abbreviations: TMAO, trimethylamine-N-oxide; FBS, Fasting Blood Sugar; HDL, high-density lipoprotein; LDL, low-density lipoprotein; ALT, alanine aminotransferase; AST, aspartate aminotransferase.

## Data Availability

The original contributions presented in this study are included in the article. Further inquiries can be directed to the corresponding author.

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
