# Peer review of "Short-Term Mediterranean Dietary Intervention Reduces Plasma Trimethylamine-N-Oxide Levels in Healthy Individuals"

_nutrients, 2025, doi:10.3390/nu17193135_

Round 1

Reviewer 1 Report

Comments and Suggestions for Authors

This study presents a timely investigation into the metabolic effects of short-term Mediterranean dietary intervention, with its clear documentation of participant selection, dietary adherence monitoring, and biochemical assessments. Its integration with existing literature on TMAO metabolism and cardiovascular risk contributes to the growing body of evidence supporting dietary modulation of gut-derived metabolites. The use of validated tools and structured dietary protocols provides a comprehensive overview of the nutritional intervention and its metabolic outcomes. However, several points merit further consideration:

1. The study primarily focuses on the reduction of plasma TMAO levels following dietary intervention. While the findings are promising, could the authors elaborate on the broader therapeutic implications and potential toxicities associated with sustained TMAO suppression? For instance, are there known physiological roles of TMAO that might be adversely affected by long-term reduction?

2. The paper suggests that TMAO reduction is independent of weight loss and primarily driven by dietary composition. Could further exploration and validation of this hypothesis—perhaps through microbiota profiling or metabolomic analyses—strengthen the mechanistic understanding of TMAO modulation and its link to dietary components?

3. Sex-specific differences in biochemical and anthropometric responses are noted, with males exhibiting more pronounced changes. It may be helpful to discuss potential physiological or behavioral factors contributing to this disparity, and whether tailored dietary strategies should be considered in future interventions.

4. The absence of a control group limits the ability to attribute observed changes solely to the Mediterranean diet. While the within-subject design is pragmatic, would the inclusion of a matched non-intervention group or crossover design enhance the internal validity and interpretability of the results?

5. The study mentions daily dietary monitoring via an online platform. Clarifying the nature and frequency of this monitoring, as well as participant compliance rates, would strengthen the methodological transparency and reproducibility of the intervention protocol.

6. Given the growing interest in gut microbiota–diet interactions, the lack of microbial profiling is a missed opportunity. Future studies incorporating microbiome analyses could provide deeper insights into individual variability in TMAO response and the role of microbial composition in mediating dietary effects.

Author Response

Dear Reviewer

First of all, I would like to thank you very much for your valuable comments.  I am most grateful to you for your invaluable feedback. I can state that all the relevant directives you have outlined have been incorporated into the document, and the requisite regulations are highlighted in the in the text.

Best regards,

Comment1: The study primarily focuses on the reduction of plasma TMAO levels following dietary intervention. While the findings are promising, could the authors elaborate on the broader therapeutic implications and potential toxicities associated with sustained TMAO suppression? For instance, are there known physiological roles of TMAO that might be adversely affected by long-term reduction?

Response1: We thank the referee for this comment. We addressed this issue and expanded our discussion in the discussion section (Page number:8,9, Lines:274-287).

Comment2: The paper suggests that TMAO reduction is independent of weight loss and primarily driven by dietary composition. Could further exploration and validation of this hypothesis—perhaps through microbiota profiling or metabolomic analyses—strengthen the mechanistic understanding of TMAO modulation and its link to dietary components?

Response2: We appreciate the reviewer's suggestion. We have referenced Li et al., 2021, and clarified that dietary composition—particularly red meat and choline intake—modulates TMAO levels independent of weight loss, with gut microbiota likely mediating these effects (Page number:9,10, Lines:334-343).

Comment3: Sex-specific differences in biochemical and anthropometric responses are noted, with males exhibiting more pronounced changes. It may be helpful to discuss potential physiological or behavioral factors contributing to this disparity, and whether tailored dietary strategies should be considered in future interventions

Response3: We thank the reviewer for highlighting this point. We have added discussion on observed sex differences in biochemical and anthropometric responses, considering both physiological and behavioral factors that may contribute to the disparities (Page number:10, Lines:346-363).

Comment4: The absence of a control group limits the ability to attribute observed changes solely to the Mediterranean diet. While the within-subject design is pragmatic, would the inclusion of a matched non-intervention group or crossover design enhance the internal validity and interpretability of the results?

Response4: We acknowledge the reviewer’s point regarding the absence of a control group. Due to practical constraints, our study employed a within-subject design without a non-intervention group. We have highlighted this as a limitation in the discussion and agree that future studies including matched control or crossover designs would provide stronger evidence for causal effects of the Mediterranean diet (Page number:11, Lines:409-413).

Comment5: The study mentions daily dietary monitoring via an online platform. Clarifying the nature and frequency of this monitoring, as well as participant compliance rates, would strengthen the methodological transparency and reproducibility of the intervention protocol.

Response5: We appreciate the suggestion. Dietary intake was monitored daily via WhatsApp, and participants received personalized feedback. Progress was tracked meal-by-meal, but reporting was based on participant self-reporting, which is considered a limitation (Page number:11, Lines:414-419).

Comment6: Given the growing interest in gut microbiota–diet interactions, the lack of microbial profiling is a missed opportunity. Future studies incorporating microbiome analyses could provide deeper insights into individual variability in TMAO response and the role of microbial composition in mediating dietary effects.

Response6: We appreciate the reviewer’s comment. As noted in the discussion, we acknowledge the absence of gut microbiota profiling as a study limitation. This study was conducted within the scope of a PhD project and was subject to budgetary constraints, which precluded microbiome analyses. We have also added that future studies incorporating microbiome analyses could provide deeper insights into individual variability in TMAO response (Page number:11, Lines:421-422).

Reviewer 2 Report

Comments and Suggestions for Authors

TMAO are supposed to contribute to cardiovascular diseases which are often associated with increased oxidative stress and chronic inflammation. 

It is shown that TMAO level is reduced in subject submitted to the Mediterranean diet. This is a good argument supporting the interest of mediterranean diet to reduce cardiovascular diseases. However, nothing is shown on parameters associated with oxidative stress and inflammation. 

This point must be presented in the introduction. 

In addition, if TMAO decrease, this could may be have an impact on oxidative stress and inflammation. Oxidative stress must be evaluated at least with MDA or conjugated dienes (this is easy to do; alternative methods are also welcome). In addition, inflammation could be presented via CRP. These data could be presented in the manuscript or as supllemenary data. This would reinforce the interest of the study and its nutrional and clinical interest.

In the discussion, the impact of TMAO on oxidative stress and inflammation must be taken in consideration.

Author Response

Dear Reviewer

First of all, I would like to thank you very much for your valuable comments.  I am most grateful to you for your invaluable feedback. I can state that all the relevant directives you have outlined have been incorporated into the document, and the requisite regulations are highlighted in the in the text.

Best regards,

Comment: TMAO are supposed to contribute to cardiovascular diseases which are often associated with increased oxidative stress and chronic inflammation. It is shown that TMAO level is reduced in subject submitted to the Mediterranean diet. This is a good argument supporting the interest of mediterranean diet to reduce cardiovascular diseases. However, nothing is shown on parameters associated with oxidative stress and inflammation. This point must be presented in the introduction. In addition, if TMAO decrease, this could may be have an impact on oxidative stress and inflammation. Oxidative stress must be evaluated at least with MDA or conjugated dienes (this is easy to do; alternative methods are also welcome). In addition, inflammation could be presented via CRP. These data could be presented in the manuscript or as supllemenary data. This would reinforce the interest of the study and its nutrional and clinical interest. In the discussion, the impact of TMAO on oxidative stress and inflammation must be taken in consideration.

Response: We thank the reviewer for the suggestion regarding oxidative stress and inflammatory markers. Unfortunately, these parameters were not assessed in the current study, as the primary focus was on TMAO levels. Nevertheless, we acknowledge the importance of this point and have added statements in both the introduction (Page number:2 Lines:59-67) and discussion (Page number:10 Lines:367-379) highlighting the potential effects of TMAO on oxidative stress and inflammation. We agree that these markers should be evaluated in future studies to provide further mechanistic insights.

Round 2

Reviewer 2 Report

Comments and Suggestions for Authors

The paper has been strongly improved and the authors answer with lot of details to the different remarks;  additionnal information were provided.

The paper is now clear, the data are sound and well discussed.